# Establishment of a quadruplex real-time PCR assay to distinguish the fungal pathogens *Diaporthe longicolla, D. caulivora, D. eres*, and *D. novem* on soybean

**Behnoush Hosseini, Ralf T. Voegele, Tobias I. Link**  *

Faculty of Agricultural Sciences, Department of Phytopathology, Institute of Phytomedicine, University of Hohenheim, Stuttgart, Germany

* tobias.link@uni-hohenheim.de

**Data Availability Statement:** All relevant data are within the paper and its Supporting Information files.

## Abstract

*Diaporthe* species are fungal plant pathogens of many important crops. Seed decay is one of the most important diseases on soybean. It is caused by various species of the genus *Diaporthe* and responsible for significant economic damage. In central Europe the four species *D. longicolla, D. caulivora, D. eres*, and *D. novem* are considered the principal species of *Diaporthe* on soybean. Fast and accurate detection of these pathogens is of utmost importance. In this study four species-specific TaqMan primer-probe sets that can be combined into a quadruplex assay were designed based on *TEF* sequences. The specificity and efficiency of the primer-probe sets were tested using PCR products and genomic DNA from pure cultures of the four *Diaporthe* species and other soybean fungal pathogens. Our results indicate that the primer-probe sets DPCL, DPCC, DPCE, and DPCN allow discrimination of *D. longicolla, D. caulivora, D. eres*, and *D. novem*, respectively, and can be used to detect and quantify these four *Diaporthe* species in parallel using quadruplex real-time PCR. In addition, the quadruplex real-time PCR assay was evaluated on different plant materials including healthy and infected soybean seeds or seed lots, soybean stems, and soybean leaves. This assay is a rapid and effective method to detect and quantify *Diaporthe* species from samples relevant for disease control.

## Introduction

Soybean (*Glycine max* L.) is one of the major sources of oil and protein in the world. Due to the expansion of soybean cultivation soybean diseases are also becoming relevant in the new soybean growing areas. Among the most important pathogens of soybean, fungal species of the genus *Diaporthe* cause severe diseases including seed decay, pod and stem blight, and stem canker [1]. Soybean seeds are heavily attacked by *D. longicolla* (Hobbs) and other species of the genus *Diaporthe*, which can significantly affect yield, quality, and stability of this industrial crop [2]. Infected soybean seeds are smaller and lighter than healthy seeds and their germination rate is reduced, which leads to their economic devaluation. Often seeds have wrinkled and

**Funding:** This research was supported by the German Federal Office for Agriculture and Food (SoySound; Grant number 2815EPS082; to TL) and the Food Security Center (FSC) of University of Hohenheim to BH. The funders had no role in study design, data collection and analysis, decision to publish, or preparation of the manuscript.

**Competing interests:** The authors have declared that no competing interests exist.

cracked seed coats covered with fungal mycelium [3]. Sometimes, however, infected seeds are symptomless. In this case, it is challenging to distinguish healthy from infected soybean seeds. Therefore, the accessibility of a fast, accurate, and sensitive method for detection and species identification for these pathogens is required. On the one hand this will help to ensure use of healthy seeds to prevent spreading of the disease. On the other hand the prevalence of the species in Germany has not been studied beyond an assessment establishing the presence of four *Diaporthe* species [4] (see below), therefore, an assay with the ability to distinguish between the species is necessary to study disease epidemiology. Future epidemiological studies should also more precisely determine how the pathogens spread, both inside plants and from plant to plant. In the seed-plating assay as a common diagnostic method, *Diaporthe* species can be identified based on their morphological characteristics by an expert mycologist. This culture-based assay is multi-stage and time-consuming. Also, because of the existence of overlaps in cultural characteristics among *Diaporthe* species, differentiation is difficult and often inaccurate [4].

These difficulties have led to the development of molecular tools to improve accuracy and reliability. Technologies based on polymerase chain reaction (PCR) enhance detection and provide comparison of *Diaporthe* spp. at the molecular level. Genetic differences among species of *Diaporthe* from different hosts have been studied by sequencing the ribosomal DNA (rDNA) internal transcribed spacer (ITS) [5]. In a study by Zhang et al. [6], specific-primers PhomI and PhomII based on the sequence of the ITS region were designed to detect *D. phaseolorum* and *D. longicolla* in soybean plants and seeds. Furthermore, PCR-restriction fragment length polymorphism (RFLP) was used to amplify DNA within the ITS region (using ITS4/ ITS5 primers) to distinguish isolates of *D. phaseolorum* and *D. longicolla* from other fungal soybean pathogens. However, a shortage of adequate polymorphisms in the ITS region made identification of other gene sequences necessary that could be used to accurately resolve the species [7]. Using translation elongation factor 1-alpha (EF1-α/TEF), beta-tubulin (TUB), histone H3 (HIS) and calmodulin (CAL) loci has promoted accurate delineation of *Diaporthe* spp. as they exhibit sufficient polymorphism [4,8–10].

Although PCR-based technologies can significantly reduce the time needed for diagnosis compared to conventional culture methods, they still require additional work [11]. Moreover, conventional PCR is not suited for quantitative analysis of plant pathogens [12].

By contrast, real-time, or quantitative PCR (qPCR) is a versatile advanced technology that allows to distinguish the target pathogen simply, quickly, economically, and reproducibly. It can also discriminate between two closely related organisms. Additionally, qPCR can be utilized to measure the pathogen load in a sample [13]. The use of probes with different fluorescent dyes enables the detection of several target pathogen DNAs in a single reaction (multiplex-PCR). The method has been used to enhance biosecurity [14] and to detect seed-borne pathogens [15], and phytopathogenic bacteria [16]. Using real-time PCR to collect epidemiological data can lead to a better understanding of seed-borne diseases. A real-time PCR assay has been developed to detect and quantify species of the genus *Diaporthe* including *D. longicolla, D. phaseolorum* var. *meridionalis, D. caulivora,* and *D. phaseolorum* var. *sojae* from soybean seeds [17]. This assay was recently applied by Kontz et al. [18] for direct quantification of *D. caulivora* and *D. longicolla* in infected soybean plants, and to quantify resistance in soybean germplasm to these two pathogens. However, the primer-probe combinations PL-3 and DPC-3 by Zhang et al. [17] were not developed for parallel use in a multiplex real-time PCR system. The specificity of these combinations for either *D. longicolla* or *D. caulivora* resides in the primers while the probes bind to DNA from either species. For use in diagnostics to test seeds for the presence of pathogens, multiplex is highly desirable, since one sample can be tested for different pathogens in a single reaction. This saves both time and costs.

We have recently surveyed *Diaporthe* spp. on soybean in central Europe [4]. In the course of this survey we established the presence of *D. longicolla*, *D. caulivora* (J.M. Santos), *D. eres* (Nitschke), and *D. novem* (J.M. Santos). We expect that if these species are allowed to spread over the expanding soybean fields in central Europe they will cause considerable damage. To provide diagnostics to prevent this we have developed and present here a multiplex (quadru-plex) real-time PCR assay to detect, distinguish, and quantify these four closely related species in parallel. The assay was tested with soybean seeds and soybean plant tissue.

## Materials and methods

### Fungal strains and plant material

Single-spore isolates of ten strains of *Diaporthe* isolated from soybean seeds ([4] or received from Kristina Petrović (Institute of Field and Vegetable Crops, Novi Sad, Serbia), respectively) were used in this study (Table 1) for DNA preparation to test the specificity of the assay.

Pure cultures of the common soybean pathogens *Sclerotinia sclerotiorum* (Lib.), *Colletotri-chum truncatum* (Schwein.), *Cercospora kikuchii* T. Matsumoto &Tomoy., (1925), and *Fusar-ium tricinctum* El-Gholl (1978) were used as controls. For *S. sclerotiorum* two isolates were used: *S. sclerotiorum* DSM 1946 (GenBank Accession: MH857810.1; DSMZ, Braunschweig, Germany) and *S. sclerotiorum* IZS (own isolate). Isolates of *C. truncatum* and *F. tricinctum* were from our laboratory collection. Additional control species were *Fusarium solani* (Mart.), two isolates of *Alternaria spp*, and three rust species: *Phakopsora pachyrhizi* (Syd.), *Uromyces fabae* (Bary ex Cooke), and *Uromyces appendiculatus* (Unger). Cultures were grown on acidi-fied potato dextrose agar (APDA) or potato dextrose agar (PDA), respectively, for 10 days at 25 ± 2˚C. Rust fungi were propagated by inoculating the respective host plants.

**Table 1. *Diaporthe* strains used in this study and their corresponding GenBank accession numbers.**

| Isolate no. | Species | GenBank Accession | |
|---|---|---|---|
| | | ITS | TEF |
| DPC_HOH1 | *D. longicolla* | MK024676 | MK099093[c] |
| DPC_HOH20[a] | | MK024695 | MK099112 |
| DPC-HOH17[b] | | MK024692 | MK099109 |
| DPC_HOH22[b] | | MK024697 | MK099114 |
| DPC_HOH25[b] | | MK024700 | MK099117 |
| DPC_HOH26[b] | | MK024701 | MK099118 |
| DPC_HOH28[a] | | MK024703 | MK099120 |
| DPC_HOH29[b] | | MK024704 | MK099121 |
| PL-157a/PDS157A[a] | | JQ697845 | JQ697858 |
| DPC_HOH2[a] | *D. caulivora* | MK024677 | MK099094[c] |
| DPC_HOH3[a] | *D. eres* | MK024678 | MK099095[c] |
| DPC_HOH7[a] | | MK024682 | MK099099[c] |
| PS-74[a] | | JF430488 | JF461474 |
| DPC_HOH8[a] | *D. novem* | MK024683 | MK099100[c] |
| DPC_HOH11[a] | | MK024686 | MK099103[c] |
| DPC_HOH15[a] | | MK024690 | MK099107 |
| DC-27(1)/17-DIA-034[a] | *D. aspalathi* | MK942646 | MK941268 |
| PS-22[a] | *D. foeniculina* | JF430495 | JF461481 |

[a]These *Diaporthe* isolates were used to test the specificity and sensitivity of the TaqMan primer-probe combinations

[b]Infected stem samples were obtained from diseased soybean plants inoculated with these *Diaporthe* isolates in greenhouse pathogenicity tests.

[c]Sequences used for primer/probe design.

Stem samples covered with pycnidia were taken from four months-old plants that had been artificially inoculated with the *D. longicolla* isolates indicated in Table 1 in greenhouse pathogenicity tests [4]. Leaf and stem samples from four-week-old healthy soybean plants were used as control.

Seed samples were taken from seed lots (cultivars Sultana and Primus) known to contain seeds infected with *Diaporthe* spp. [4]. More infected soybean seeds (with and without symptoms; cultivar Anuschka) were from Landwirtschaftsbetrieb Zschoche (Südliches Anhalt, Germany). Healthy soybean seeds (healthy as determined by the source, no symptomatic seeds in the sample; cultivar Sultana) were obtained from the Landwirtschaftliches Technologiezentrum (LTZ) Augustenberg (Karlsruhe, Germany).

## DNA extraction from mycelia

Mycelia were scraped from ten-day-old fungal cultures on APDA plates and homogenized by vortexing for 40 s using microbeads (Lysing Matrix E tubes, Fast Prep-24™, MP Biomedicals GmbH, Eschwege, Germany) in lysis buffer. DNA from *Diaporthe* strains was prepared using the peqGOLD fungal DNA Mini Kit (PEQLAB Biotechnologie GmbH, Erlangen, Germany) following the recommendations of the manufacturer. DNA from other soybean pathogens was isolated using the protocol by Liu et al. [19]. DNA concentrations were determined by measuring the absorption at 260 nm.

## DNA extraction from plant material

Prior to DNA extraction, stem samples (each 2 cm) were ground individually in liquid nitrogen for 2 min using mortar and pestle. Leaf samples ($\leq$ 100 mg) were placed individually into 2 ml micro screw tubes (Sarstedt, Nümbrecht, Germany) together with two steel balls (4.50 mm, Niro, Sturm Präzision GmbH, Oberndorf am Neckar, Germany). They were frozen for 3 min in liquid nitrogen, and homogenized for 20 s using the FastPrep®-24 homogenizer (MP Biomedicals GmbH). The micro screw tubes were returned to liquid nitrogen for 1 min, and homogenization was repeated two more times.

For DNA extraction from seeds, soybean seeds were treated with 1% sodium hypochlorite for 30 s, followed by washing with sterile distilled water. Surface-disinfected soybean seeds were squeezed to remove their seed coats. Seed coats were placed individually in 2 ml micro screw tubes containing 2 steel balls, frozen for 3 min in liquid nitrogen, and homogenized using the FastPrep®-24 homogenizer as described above. Uncoated soybean seeds and whole seeds were ground individually in liquid nitrogen by using mortar and pestle for three minutes. Seeds were each treated individually as separate samples.

The extraction of DNA for all homogenized plant material was done using the DNAeasy Plant Mini Kit (Qiagen, Hilden, Germany), following the manufacturer's instructions.

## Design of TaqMan primer-probe sets

Multiple sequence alignments were performed using ClustalW as implemented in BioEdit (version 7.1.3.0 [20]). Melting temperatures (Tm) and potential secondary structures of primers and probes were evaluated with Gene Runner (Version 6.5.52x64 Beta).

Specificity of the selected primers was tested using NCBI's Primer-BLAST (https://www.ncbi.nlm.nih.gov/tools/primer-blast/). In "Primer Pair Specificity Checking Parameters" we entered nr as database and as organisms *Diaporthe*, *Fungi*, *Pythium*, *Phytophthora*, and *Glycine*. All oligonucleotide primers and probes were synthesized by Biomers.net GmbH (Ulm, Germany) and are shown in Table 2.

**Table 2. TaqMan primer-probe combinations for detection and distinguishing *D. longicolla*, *D. caulivora*, *D. eres* and *D. novem*.**

| Primer-probe set/ Specificity | Primer Probe | Sequence | Target isolate | Position (bp) | Fragment length (bp) | Efficiency (%) | |
|---|---|---|---|---|---|---|---|
| | | | | | | PCR product | Genomic DNA |
| DPCL/DL | DPCL-F | 5′-TGTCGCACCTTTACCACTG-3′ | DPC-HOH20 | 199–217 | 90 | 98.2 | 81.0 |
| | DPCL-R | 5′-GAACGATCCAAAAAGCTCTC-3′ | | 269–288 | | | |
| | DPCL-P | FAM-GCATCACTTTCATTCCCACTTTCTG-BMN-Q535 | | 239–263 | | | |
| DPCC/DC | DPCC-F | 5′-GCCTGCAAAACCCTGTTAC-3′ | DPC-HOH2 | 186–204 | 120 | 97.7 | 90 |
| | DPCC-R | 5′-CATCATGCTTTAAAAATGGGG-3′ | | 285–305 | | | |
| | DPCC-P | Cy5-CTCTTACCACACCTGCCGTCG-BMN-Q620 | | 237–257 | | | |
| DPCE/DE | DPCE-F | 5'-ACTCACTCAATCCTTGTCAC-3' | DPC-HOH3 | 208–227 | 100 | 82.4 | 92.2 |
| | | | | 288–307 | | | |
| | DPCE-R | 5'-GAGGGTCAGCATAATATTCG 3' | | 244–266 | | | |
| | | | DPC-HOH7 | 208–227 | 101 | | |
| | DPCE-P | ROX-CCATCAACCCCATCGCCTCTTTC-BMN-Q590 | | 289–308 | | | |
| | | | | 245–267 | | | |
| DPCN/DN | DPCN-F | 5′-AAAACCCTGCTGGCATTAAC-3′ | DPC-HOH8 | 192–211 | 99 | 94.5 | 93 |
| | | | | 270–290 | | | |
| | DPCN-R | 5′-TATTCTTGACAGTTCGTTTCG-3′ | | 238–262 | | | |
| | | | DPC-HOH11 | 193–212 | 99 | 95.5 | 84.2 |
| | DPCN-P | HEX-TCTACCACTTTCAACCCTATCAATC-BMN-Q535 | | 271–291 | | | |
| | | | | 239–263 | | | |

F, R, P = Forward, Reverse, and Probe. The probes carry ROX, FAM, Cy5, or HEX dyes as reporter attached to the 5′-terminal nucleotide and BMN-Q590, BMN-Q535, or BMN-Q620 as quencher attached to the respective 3′-terminal nucleotide.

DE = *D. eres*, DL = *D. longicolla*, DC = *D. caulivora*, and DN = *D. novem*.

## Real-time PCR

Real-time PCR was performed using a CFX96 Real-Time PCR system (Bio-Rad Laboratories GmbH, Munich, Germany) using FrameStar® 96-Well Skirted PCR Plates (4titude, Brooks Automation, Chelmsford, MA, USA). Real-time PCR reactions were prepared using a ready to use mixture, SensiFAST Probe No-ROX mix (2x) (SensiFAST™ Probe No-ROX Kit, Bioline GmbH, London, UK). All reactions were performed with a final volume of 20 μl. Reaction mixtures in singleplex real-time PCR assays consisted of 10 μl 2x SensiFAST Probe No-ROX mix, 8 pmol of each forward and reverse primers, 2 pmol probe, and 2 μl template DNA. Reaction mixtures for duplex and quadruplex real-time PCR assays consisted of 10 μl 2x SensiFAST Probe No-ROX mix, and a reduced amount of 4 pmol of each forward and reverse primers, 1 pmol of each of the two probes, and 2 μl template DNA. Reaction mixtures for quantifying soybean DNA consisted of 10 μl 2x SensiFAST SYBR No-ROX mix, 8 pmol each of primers GmUKN2f (5'-GCCTCTGGATACCTGCTCAAG-3') and GmUKN2r (5'-ACCTCCTCCTCAAACTCCTCTG-3') [21], and 2 μl template DNA.

Samples were incubated for 3 min at 95°C and then subjected to 40 cycles of 95°C for 15 s and 60°C for 45 s. Reactions used as standards were run in technical triplicates; reactions used to test for pathogen presence were run in technical duplicates. No-template controls were included for all assays, singleplex, duplex and quadruplex.

## Dilution series of PCR products and genomic DNA

TEF regions were amplified via PCR using primers EF1-728F (5'-CATCGAGAAGTTCGAGAAGG-3') and EF1-986R (5'-TACTTGAAGGAACCCTTACC-3') [22] in individual reactions for

the different *Diaporthe* isolates. Amplification was performed in 25 µl reactions: 2.5 µl 10x Taq buffer with $(NH_4)_2SO_4$ (Thermo Fisher Scientific, Waltham, MA, USA), 2.5 µl 2 mM dNTPs, 2.5 µl $MgCl_2$ (25 mM), 12.5 pmol of each forward and reverse primers, 1 µl Taq DNA polymerase (1 U/µl), and 1 µl genomic DNA. The following conditions were used: 3 min at 95˚C, 35 cycles of: denaturation 30 s at 95˚C, annealing 30 s at 68˚C, and elongation 30 s at 72˚C, and a final elongation of 5 min at 72˚C. PCR products were checked by electrophoresis on 2% agarose gels. Subsequently, PCR products were purified using the PEQGOLD Cycle-Pure Kit (PEQLAB Biotechnologie GmbH). DNA concentrations were determined by using a Qubit® 2.0 Fluorometer (Thermo Fisher Scientific). To determine the amplification efficiency of each primer-probe set, serial dilutions of PCR products containing $10^9$ to $10^4$ copies/µl and also dilution series 1:10, 1:100 and 1:1000 for the genomic DNA of the *Diaporthe* isolates were prepared. To allow for quantification standard curves were performed with genomic DNA of the *Diaporthe* isolates. For the latter the concentration of genomic DNA was both determined by measuring absorption at 260 nm and by using a Qubit® 2.0. The DNA was diluted 1:10–1:$10^6$ with 50 µg/ml DNA prepared from healthy soybean tissue.

## Results

### Design of TaqMan primer-probe sets for specific detection of *D. longicolla*, *D. caulivora*, *D. eres*, and *D. novem* in a multiplex real-time PCR

With the aim of building a quadruplex real-time PCR assay to detect *D. longicolla*, *D. caulivora*, *D. eres*, and *D. novem*, the *Diaporthe* species we previously identified in central European soybean seeds [4], we first checked the literature for primer-probe sets that might be used. Prominent among the studies addressing this issue was the one by Zhang et al. [17]. After establishing that the primer-probe sets developed by Zhang et al. [17] could not be used together in a multiplex reaction we designed our own primer-probe sets by searching alignments of *Diaporthe* sequences for suitable sites. We started with alignments containing the sequences of 32 *Diaporthe* isolates [4] together with sequences of ex-type species and removed identical sequences to gain alignments with the fewest possible number of sequences representing the sequence diversity. From the ITS alignment it became clear that it is not possible to design sets in this region because the sequences are too similar. Therefore, all primer-probe sets were designed *de novo* in the TEF alignment (Fig 1, Table 2).

Primers were checked for specificity using Primer-BLAST. It could be corroborated that the primer-probe combinations can distinguish between the four *Diaporthe* species and do not detect any other pathogens occurring on soybean in Central Europe (S1 Text).

### Assessing the specificity and efficiency of the TaqMan primer-probe sets

**Singleplex reactions.** In order to test primer efficiency we used serial dilutions of PCR products. This way, problems with inhibitors can be avoided and the minimum template copy

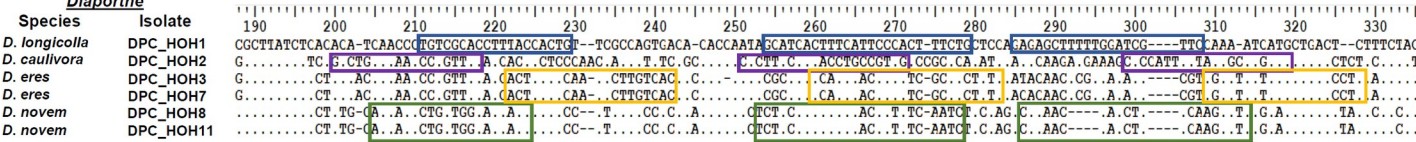

**Fig 1. Primer and probe specificity based on alignment.** Position of primer-probe combinations DPCL (blue frames), DPCC (purple frames), DPCE (orange frames), and DPCN (green frames) in the alignment of TEF sequences of *Diaporthe* species. Identical nucleotides and gaps are represented by dots and dashes, respectively. The sequences in the alignment were chosen from the sequences obtained in [4] in order to represent all intra specific polymorphisms, identical sequences were removed.

number can be extrapolated. To test conditions corresponding to what will be used in the screen for *Diaporthe* infestations genomic DNAs of the *Diaporthe* isolates were also tested. We obtained acceptable efficiencies with the PCR products and with the genomic DNAs (S1 Fig, Table 2). From our dilution series with PCR products we could also deduce that we can detect as few as ten copies or less. Each primer probe set was also tested with genomic DNA from the non-target *Diaporthe* species (S2 Fig). No amplification was recorded with non-target species indicating good specificity of the primer probe sets.

**Multiplex reactions.**  To ensure adequate efficiencies of the TaqMan primer-probe sets in the presence of other oligonucleotides and fluorogenic dyes, duplex real-time PCR assays were performed. We tested all six combinations of primer-probe sets. Each combination was tested by applying parallel dilution series with DNA of both species. The efficiencies were still good. A full description of the experiment can be found in S2 Text.

Our first step in assessing the quadruplex assay was to test again for specificity. DNA from the different isolates was added individually to the reactions. In all cases, we obtained a signal from the specific reporting dye only (Fig 2). For *D. longicolla* and *D. eres* additional isolates were tested with identical results. This proves that each primer-probe set amplifies DNA from its target species but not from the other three species. Two non-target *Diaporthe spp.*, nine other soybean pathogens, and two additional rust fungal species tested negative. Similarly, DNA extracted from healthy soybean leaves and stems was not amplified (Fig 2). This indicates high specificity of our primer-probe sets also in a quadruplex assay.

When two or three different DNA samples of species of *Diaporthe* were tested together, all four TaqMan primer-probe sets retained similar specificity and showed discrimination between the particular pathogens present in the quadruplex assay. Finally, when applying DNA of all four species of *Diaporthe* together in one PCR reaction, the appearance of signals related to all four reporting dyes proved the ability of the assay to detect *D. longicolla, D. caulivora, D. eres* and *D. novem* in parallel (Fig 3).

## Standard curves for quantification using the quadruplex real-time PCR assay

After testing the efficiency and the specificity of the primer-probe sets and after establishing that they can be used together in a quadruplex assay, we started to evaluate the full assay for its sensitivity. For this, dilution series were created with DNA from four representative isolates, using roughly 20 ng, 2 ng, 200 pg, 20 pg, 2 pg, 0.2 pg, and 0.02 pg DNA per reaction and diluting with DNA prepared from soybean tissue. The resulting standard curves (Fig 4, Table 3) can be used to calculate the amount of *Diaporthe* DNA in ng per reaction.

The full series of tests for determining the limit of detection (LOD) and a $C_q$ cutoff could not be performed so far. Nevertheless, results from our dilution series allow for a rough estimate of these values. These estimates are presented in Table 3.

## Validation of the quadruplex real-time PCR assay

**Infected soybean stems.**  For our multiplex real-time PCR to be useful it is necessary that it is capable to detect DNA from *Diaporthe* spp. prepared from infected plant samples. To validate this, we first used stem samples covered with pycnidia. These were taken from plants that had been artificially inoculated with *D. longicolla* isolates (Table 1) in greenhouse pathogenicity tests [4].

*D. longicolla* DNA was detected in all tested samples with visible symptoms (Fig 5). For healthy stem samples the primer-probe sets produced no amplification (Fig 5).

**Screening soybean seeds.**  Because *Diaporthe* spp. are seedborne pathogens, the most important application for our newly developed assay is the screening of soybean seed-lots.

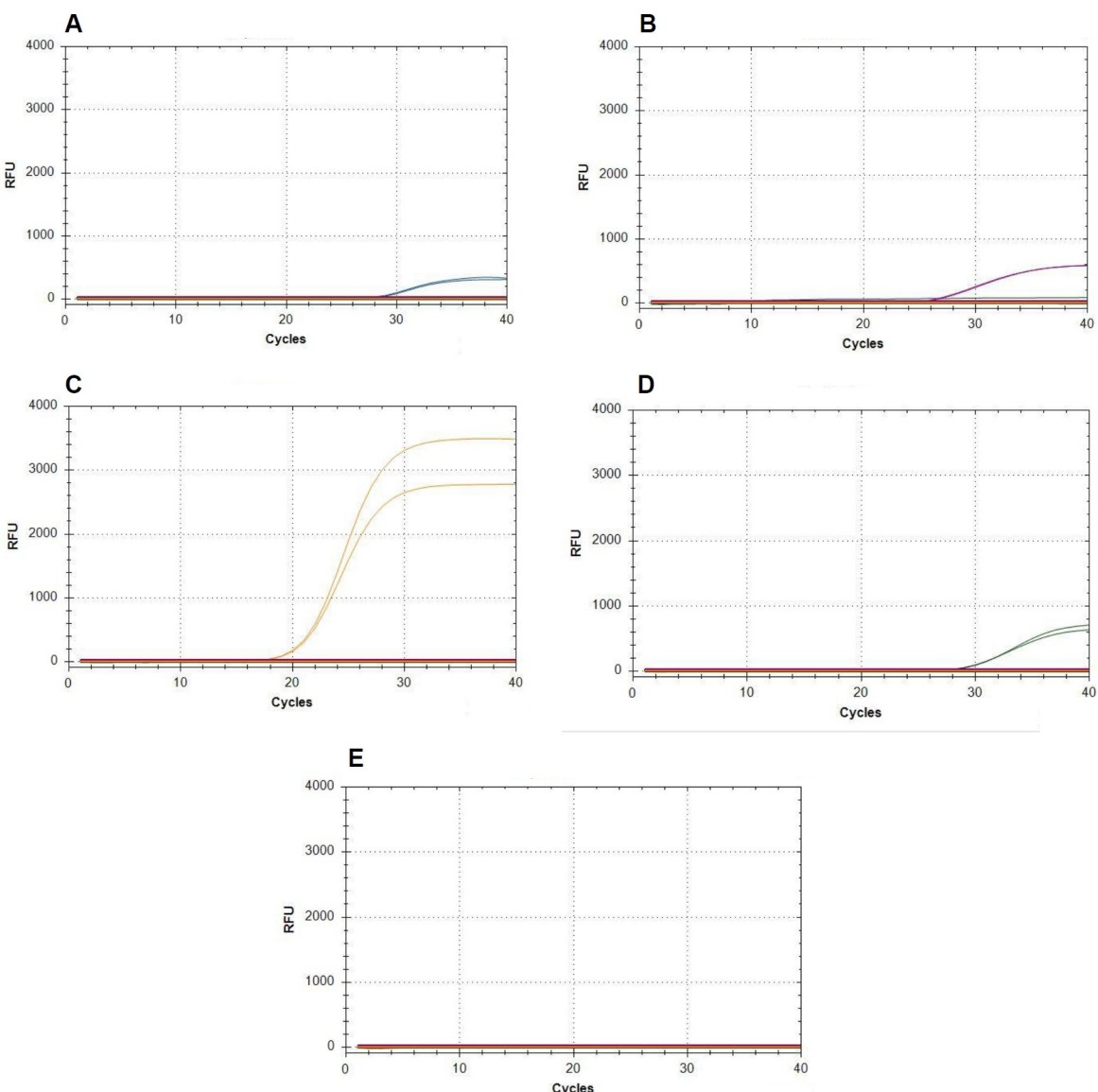

**Fig 2. Specificity of the quadruplex real-time PCR assay.** Since the graphs for different isolates of the target species and also of all the non-target species are highly similar, only one representative graph is shown each. 0.4 ng DNA from (A) *D. longicolla* DPC_HOH28, (B) *D. caulivora* DPC_HOH2, (C) *D. eres* DPC_HOH3, and (D) *D. novem* DPC_HOH15 was added individually to the mix that contained all four primer-probe sets. (E) shows the result for the non-target species *D. aspalathi*, *D. foeniculina*, *Cercospora kikuchii*, *Fusarium solani*, *Alternaria sp.*, *S. sclerotiorum* DSMZ, or *S. sclerotiorum* IZS, *Colletotrichum truncatum*, *F. tricinctum*, *Phakopsora pachyrhizi*, *Uromyces fabae*, *Uromyces appendiculatus*, healthy soybean leaf, and healthy soybean stem. For these species and also *D. longicolla* isolate PL-157a and *D. eres* isolate PS-74 DNA amounts varied between 350 and 2,500 ng.

Therefore, we tested detection of these pathogens using DNA prepared from soybean seeds. We used seed samples from seed-lots that were known to contain seeds infected with *Diaporthe* spp. and that already had been used in our earlier study [4] on identification of *Diaporthe* strains. All four *Diaporthe* species could be detected in different seed samples or seed lots. We also tested DNA preparation from whole seeds, seed coats and uncoated seeds. S3 Fig shows an example where we detected *D. eres* and *D. novem* in extracted DNA of seed coats, uncoated seeds and whole seeds via the quadruplex real-time PCR assays. For healthy seeds the primer-probe sets produced no amplification. These experiments also yielded the perception

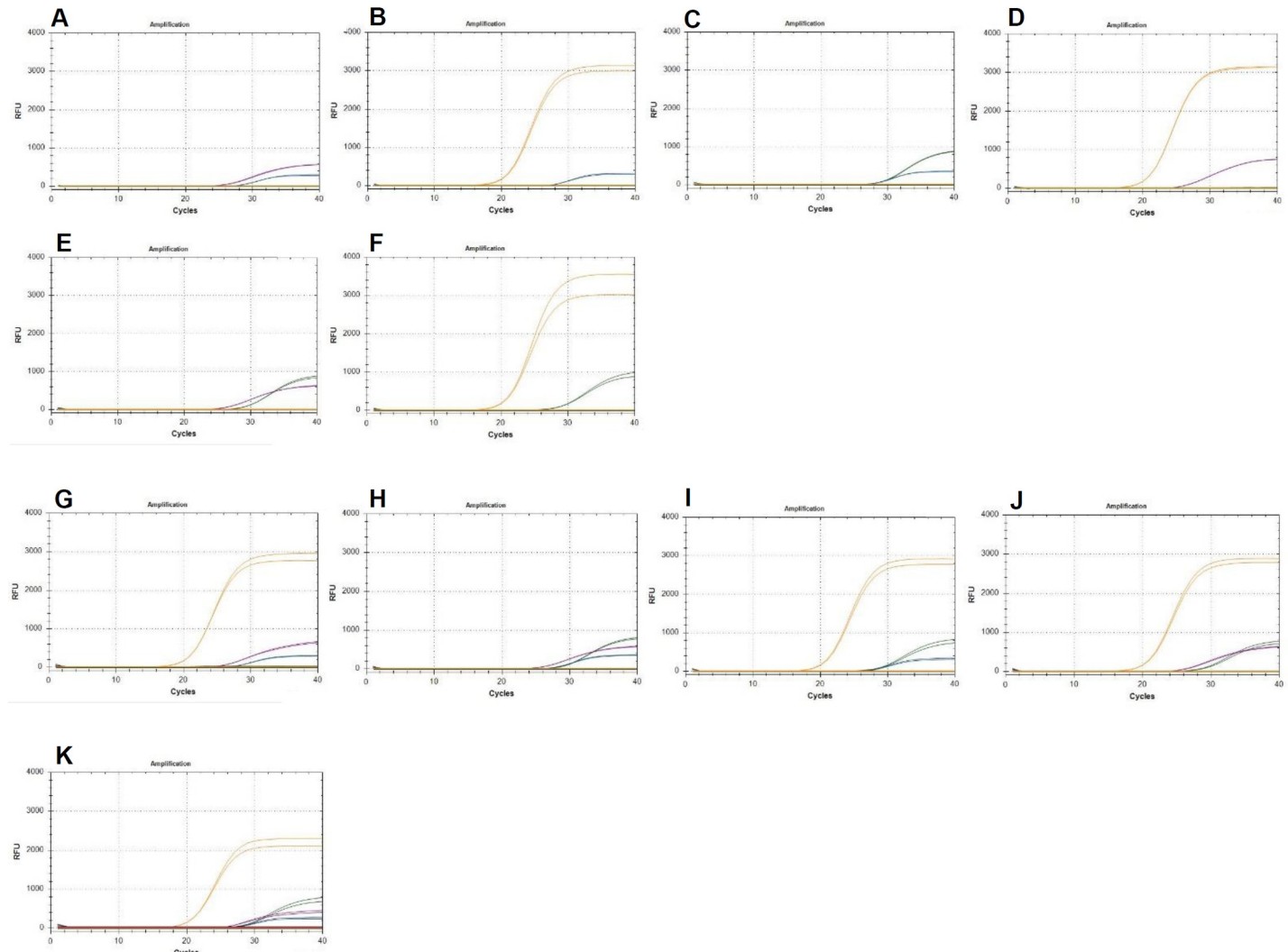

**Fig 3.** Parallel detection of two (A-F), three (G-J), or all four (K) different *Diaporthe* species. 0.4 ng DNA from (A) *D. longicolla* (blue) and *D. caulivora* (purple), (B) *D. longicolla* and *D. eres* (orange), (C) *D. longicolla* and *D. novem* (green), (D) *D. caulivora* and *D. eres,* (E) *D. caulivora* and *D. novem,* (F) *D. eres* and *D. novem,* (G) *D. longicolla, D. caulivora,* and *D. eres,* (H) *D. longicolla, D. caulivora,* and *D. novem,* (I) *D. longicolla, D. eres,* and *D. novem,* (J) *D. caulivora, D. novem,* and *D. eres,* and (K) *D. longicolla, D. caulivora, D. eres,* and *D. novem* were added to the mix that contained all four primer-probe sets.

that DNA preparation from seed coats is most easily accomplished, especially homogenization. Therefore, in the following experiments we used DNA from seed coats, only.

Finally, we present results from sampling two different seed lots (Fig 6). For each lot we prepared DNA from 30 seed coats and tested these DNA samples in the quadruplex qPCR assay. In addition, we determined the amount of soybean DNA using soybean primers in a SYBR green based qPCR reaction. The amount was calculated using the standard curve for soybean DNA ($C_q = 30.9-3.6x$). Thus, we can quantify the severity of an infection in ng pathogen DNA/ng soybean DNA.

It is apparent that in any seed lot there are seeds infected with *Diaporthe spp.* while other seeds seem to be free of the pathogens. On the other hand, the portion of fungal DNA that can be found in a seed varies by more than an order of magnitude (pg per ng soybean DNA up to ng per ng soybean DNA). For establishing the assay in the seed certification process it still needs to be established whether the portion of fungal DNA in different seeds should be taken

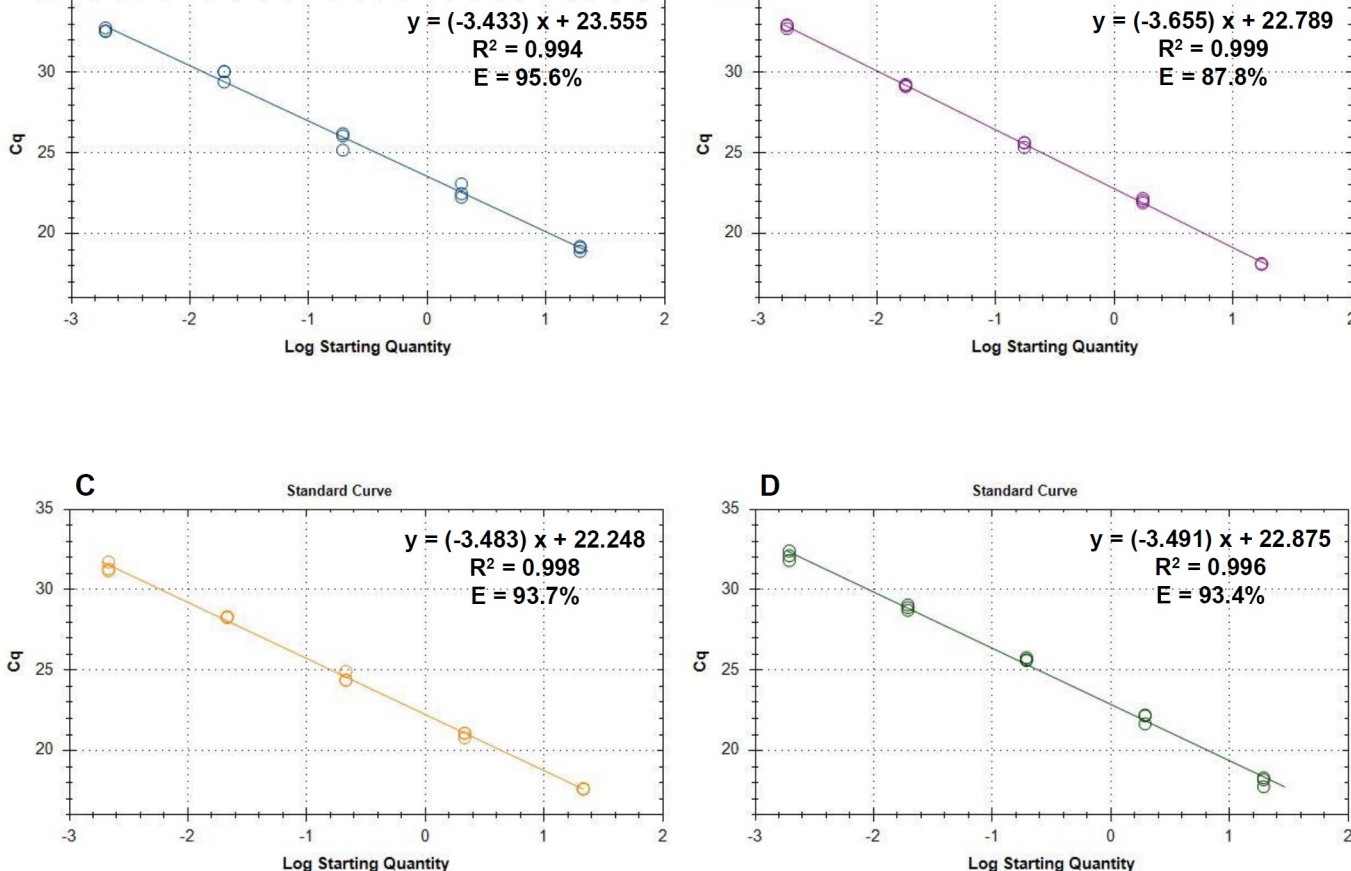

**Fig 4. Standard curves for quantification of *Diaporthe spp*. using the quadruplex qPCR assay.** (A) Graph and data of the standard curve for *D. longicolla*. The highest starting amount was 19.4 ng. The fluorescence threshold was set at 10 RFU. (B) Graph and data of the standard curve for *D. caulivora*. The highest starting amount was 17.4 ng. The fluorescence threshold was set at 35 RFU. (C) Standard curve for *D. eres*. The highest starting amount was 21.4 ng. The fluorescence threshold was set at 84 RFU. (D) Standard curve for *D. novem*. The highest stating amount was 19.4 ng. The fluorescence threshold was set at 42 RFU.

**Table 3. Functions and additional information derived from standard curves.**

| Species | Isolate[a] | Function[b] [$C_q$] | LOD[c] [pg] | $C_q$ cutoff[d] |
|---|---|---|---|---|
| *D. longicolla* | DPC_HOH20 | = 23.6–3.4x | 0.2 > X > 0.02 | 36 > X > 39 |
| *D. caulivora* | DPC_HOH2 | = 22.8–3.5x | 0.2 > X > 0.02 | 35 > X > 38 |
| *D. eres* | DPC_HOH7 | = 22.2–3.5x | 0.2 > X > 0.02 | 33 > X > 36 |
| *D. novem* | DPC_HOH11 | = 22.7–3.3x | 2[c] > X > 0.02 | 32 > X >37 |

[a]Isolate from which DNA was prepared for the standard curve experiment.

[b]Function describing the standard curve. x = log10 starting quantity in ng.

[c]Estimate for the limit of detection showing the DNA amount from the standard curve experiment that still gave an amplification and the first amount that did not give amplification. For *D. novem* one of the reactions at 0.2 ng was also negative; this is responsible for the very wide range in this case.

[d]Estimate for the $C_q$ cutoff derived from the LOD: $C_q$ corresponding to the amount still giving amplification and the calculated virtual $C_q$ where no amplification was seen.

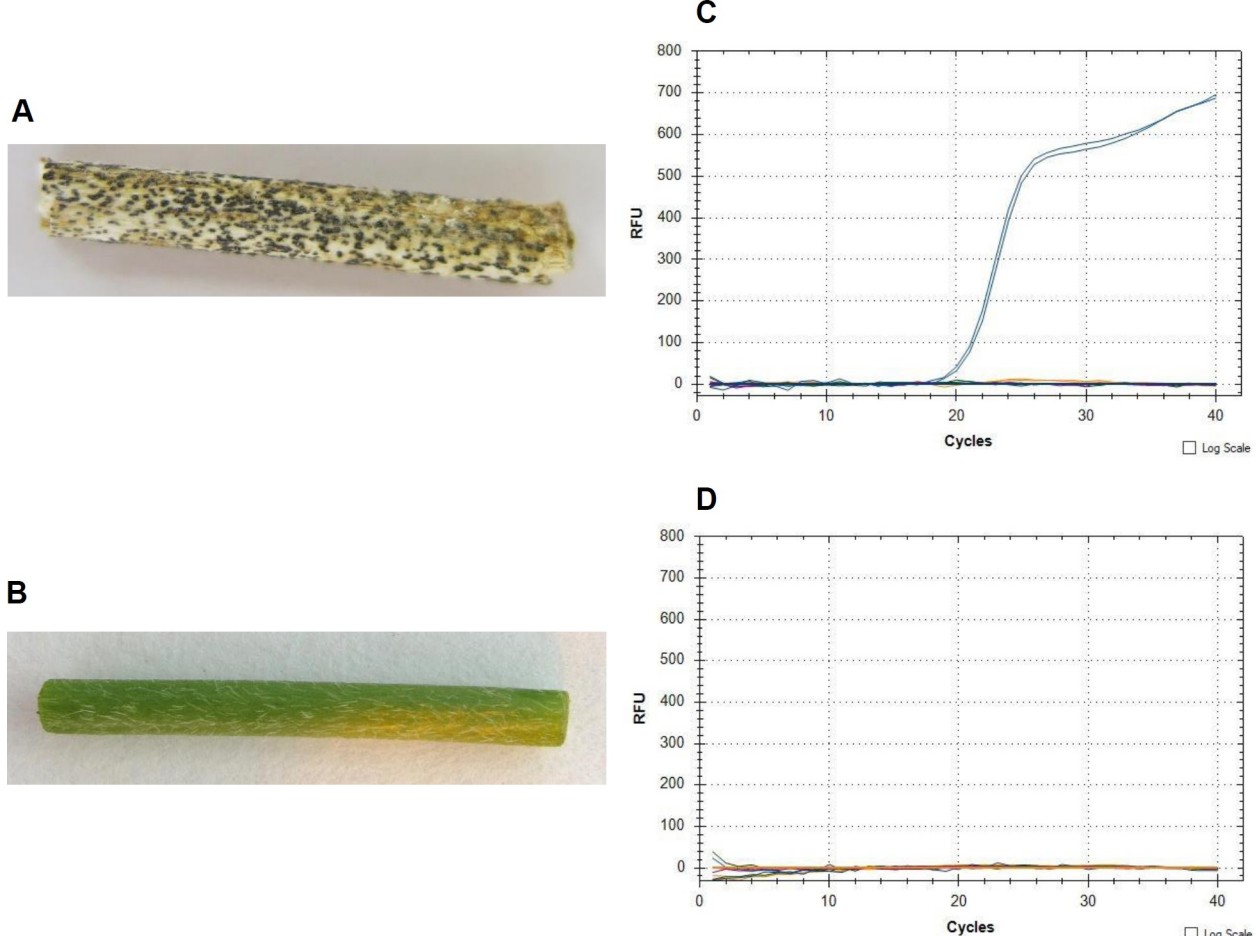

**Fig 5. Validation of the quadruplex real-time PCR assay.** (A) Soybean stem sample inoculated with *D. longicolla* isolate DPC-HOH26, (B) healthy soybean stem sample, (C) and (D) quadruplex real-time PCR on the stem samples shown in (A) and (B), respectively.

into account or whether the assay should deliver pure yes or no results for a given seed. For the latter decision a more precise determination of LOD and $C_q$ cutoff will be necessary.

## Discussion

Soybean production in central Europe has been on a very small scale, so far. This is mostly due to the cold climatic conditions. Together with smaller plot sizes this contributed to the fact that soybean production in central Europe was not competitive compared to production in the USA or South America. These conditions are now changing. There is growing demand for soybean for human consumption that is not genetically modified (non-GM soybean). In addition, there is growing reservation against importing GM soybean for animal feed. Together with warmer and drier weather caused by climate change and the introduction of new cultivars that tolerate the weather in central Europe, the conditions for growing soybean in central Europe have much improved [23].

Soybean production in central Europe is expanding rapidly, but starting from almost nothing [23]. It can be surmised that pathogens affecting soybean in other regions of the world will become important in central Europe, too. Nevertheless this assumption needs to be confirmed. As part of a larger project surveying the whole complement of pathogens on soybean in central

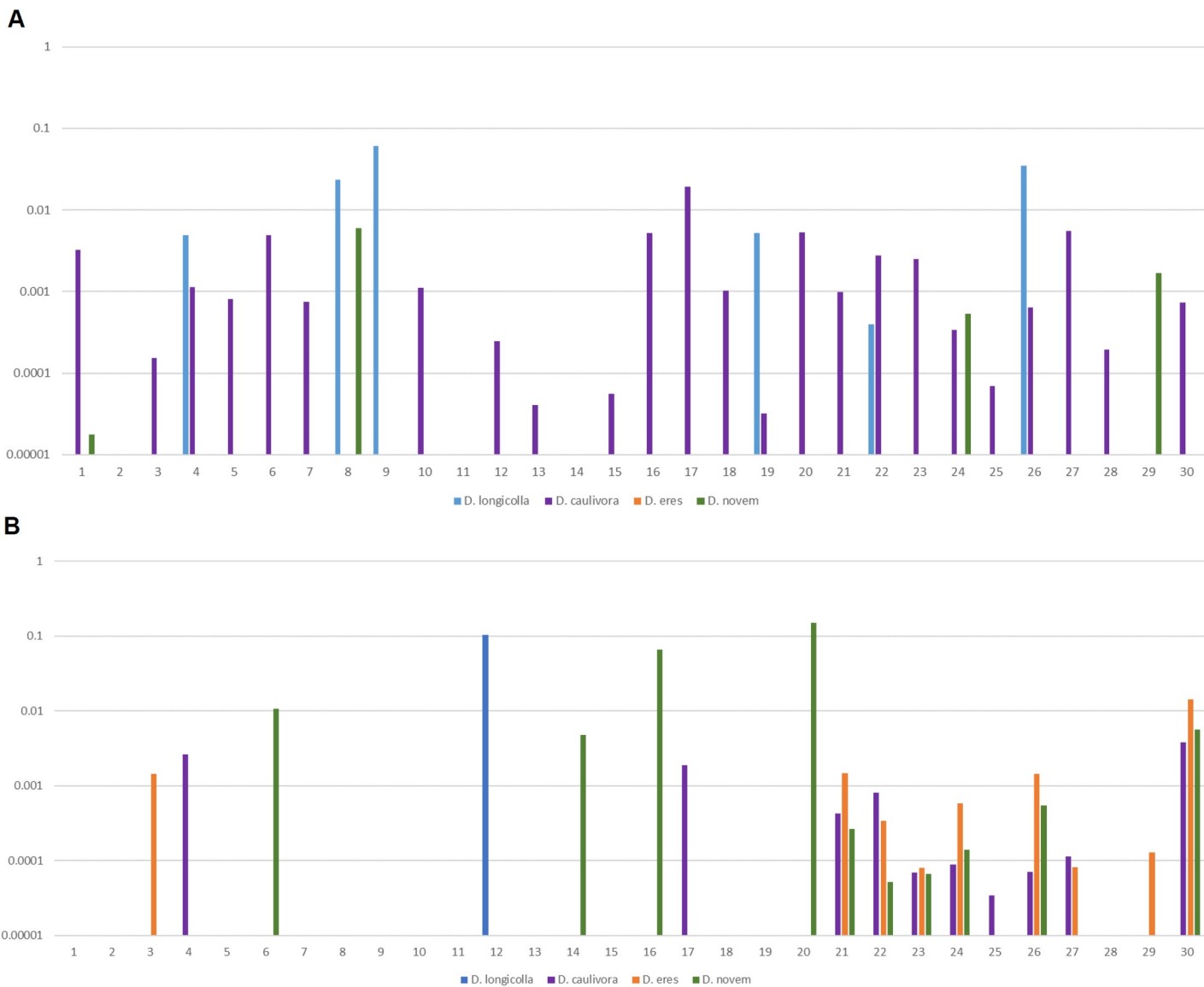

**Fig 6. Sampling soybean seed lots via the quadruplex real-time PCR assay.** (A) First seed lot, from a field in Oberweiden, Austria. (B) Second seed lot, from a field in Ebergassing, Austria. Bars represent ng *Diaporthe* DNA/ng soybean DNA; because of the strong variation a logarithmic scale was chosen. The numbers on the x-axis represent the 30 seeds that were individually tested.

Europe we have analyzed the DPC on soybean. In the course of this survey we established the presence of *D. longicolla, D. caulivora* (J.M Santos), *D. eres* (Nitschke), and *D. novem* (J.M. Santos) in central Europe [4]. In comparison, in Serbia also *D. foeniculina, D. rudis* [24], and *D. sojae* [25] were found on soybean seeds. The latter study [25], however, could not find *D. foeniculina* and *D. rudis* but determined *D. longicolla* as the dominating *Diaporthe* species followed by *D. caulivora.* This also indicates that *D foeniculina* and *D. rudis* only have marginal occurrence, which is probably also true for *D. sojae* in central Europe.

A severe limitation to these studies is the relatively small number of seeds that were tested and the much smaller number of fungal isolates that were assigned to a species. So far the rating of *D. longicolla* as the dominant species in central Europe relies on the assignation of 32 isolates [4]. To enable much wider sampling that could lead to epidemiological studies and

monitoring of the species over several years and to limit the spread of these pathogens in central European soybean fields, especially such fields newly introduced to soybean production, we set out to build a diagnostic assay for these four species.

We chose qPCR for our assay since it allows quick and specific identification of pathogens and, using TaqMan probes with different fluorescent dyes, also allows parallel detection of different species in a multiplex reaction [11]. Parallel detection of pathogens is desirable because it reduces the number of necessary qPCR reactions by the multiplexing factor. Especially with large sample numbers this can save a lot of money. On the other hand, multiplex assays are more difficult to design than singleplex reactions. In singleplex reactions the probe just adds to the specificity of the assay, which is determined by primers and probe together. In this case it can be sufficient if one of the three oligos of the primer-probe set is specific. In a multiplex, however, a probe must be specific to just one species, otherwise there is cross-detection of the different species. The primers should also be specific, because false amplifications could lead to reduced efficiencies of the assay. This means that for a multiplex all three oligos of the primer-probe set should be specific and in the case of a quadruplex that twelve unique oligonucleotides have to be identified.

Our approach was aim-oriented rather than searching for novelty and so, instead of just designing primers *de novo* we first searched the literature for existing PCR assays for detection of the four *Diaporthe* species. Prominent here was the study by Zhang et al. [17] who developed three primer-probe sets, PL-5, PL-3, and DPC-3. While PL-5 detects *D. longicolla*, *D. caulivora*, *D. phaseolorum* var. *sojae*, and *D. phaseolorum* var. *meridionalis* in parallel, PL-3 and DPC-3 are specific for *D. longicolla* and *D. caulivora*, respectively. Unfortunately, the primer-probe sets are only specific when used in singleplex reactions. Importantly the probes PL-3 P and DPC-3 P bind to the same sequence and cannot be used together in a multiplex assay.

We next made an effort to alter the primer-probe sets PL-3 and DPC-3 [17] to enable their use in a multiplex assay and to design primers and probe for *D. eres* based on the ITS sequence. However, it soon became apparent that there is too little sequence divergence between the four species within the ITS region to design primer-probe sets that are not only specific for the different species but can also be used together in a quadruplex assay. Therefore, we decided to design primers *de novo* using our alignment of TEF sequences of the four *Diaporthe* species that shows more sequence divergence (Fig 1).

Subsequently, all four TaqMan primer-probe sets were tested in singleplex assays individually to amplify the PCR products and genomic DNA of the respective *Diaporthe* strains. The four TaqMan primer-probe sets showed excellent discrimination of the sequences for which they were designed. Also, the performance of the primer-probe sets in duplex assays revealed that the efficiency of the primer-probe sets does not change markedly in the presence of a second primer-probe set and that presence of one pathogen in abundance does not mask detection of another less abundant pathogen. Furthermore, the four amplification curves in quadruplex real-time PCR showed that each primer-probe set amplifies a single product for its target species, confirming the specificity of the primer-probe sets. The accuracy of this multiplex assay was also tested with DNA from other important soybean pathogens, healthy stem and leaf tissue as negative controls, without any false positive results.

For the assay to reflect the actual pathogen load in an infected tissue, it is necessary that quantification is possible. To enable this, we created standard curves with genomic DNA prepared from pure cultures of isolates of all four *Diaporthe* species relevant in our assay. The DNA was diluted in soybean DNA to simulate tests in which DNA is prepared from soybean tissue to diagnose *Diaporthe*. With our standard curves it is now possible to quantify the four *Diaporthe* species and even though the exact LOD for the four species in the assay was not yet determined it could be deduced that it is less than 0.2 pg.

The suitability of this assay for detection of *Diaporthe* species in infected plant material was supported via detection of *D. longicolla* from stem samples inoculated with this species. There was a 100% correlation between re-identification of *D. longicolla* from stem tissue using the real-time PCR assay and results obtained using culture-based and sequencing based methods [4].

Because of the importance of seed health testing to detect seed-borne pathogens as the first step in the management of crop diseases, conventional seed detection assays including visual examination, selective media, serological assays and the seedling grow-out assay have been used extensively, but all have shortcomings ranging from inefficiency to lack of specificity and sensitivity [26]. Therefore, the multiplex real-time PCR assay was evaluated for screening of seeds. When applying extracted DNA from three different parts of infected soybean seeds to the assay, *D. eres* and *D. novem* were detected first. These first tests also showed, that when a seed is infected, the pathogen can also be found in the seed coat. Since homogenization of seed coats is easier than homogenization of whole seeds we decided that testing of seed coats is the way to proceed.

We finally applied our assay to the testing of seed lots instead of individual seeds. Even though this was the first test of the assay it shows more about the tested seed lots than what could have been learned by extensive seedling grow-out assays combined with species determination (for example [4,24,25]). Not only could we determine what was the dominant species (*D. caulivora* in the first and *D. novem* in the second seed lot), we could also detect double or even triple infections of individual seeds, that would most likely have been missed with a different assay. The main aim of the experiment was to gather data that should later help with developing sampling schemes for the assessment of seed lots. Our data show, that even in heavily infested lots as chosen for these tests, individual seeds may be or may not be infected, suggesting that it will be necessary to test a considerable number of seeds per lot before making a decision on the suitability of a given lot. We also found that the pathogen load per seed can vary considerably. It will be necessary to gather more data, to perform additional experiments studying disease progression, and to discuss with both seed companies and authorities involved in seed testing and certification to decide on the best sampling scheme and whether the pathogen load per seed should be considered in the decision on a seed lot. Seed soaking assays [27] could be an alternative to random sampling of seeds with DNA preparation from the seeds and will be considered in further tests of our assay.

In conclusion, our assay eliminates the need to obtain cultures of the pathogens for identification. It provides a rapid and practical method to detect four important and common species of *Diaporthe* directly in diseased plant tissues and infected soybean seeds. The application of our assay offers the potential to improve laboratory diagnosis of *Diaporthe* spp. in soybean seeds. It can contribute to survey the distribution of *Diaporthe* spp. in different regions, different years, and different cultivars. It could be of value for inspection of soybean seed lots and can be used to prevent the transmission of pathogens and improve disease control decision making.

## Supporting information

**S1 Fig. Standard curves to determine the efficiency of the primer probe sets DPCL, DPCC, DPCE, and DPCN in singleplex reactions.** Graphs showing the quantification cycle ($C_q$) on the y-axis and the quantity of TEF PCR product (10 to $10^{10}$ copies) ((A), (C), (E), (G), (I)) or genomic DNA (undiluted, 1:10, 1:100 and 1:1,000) ((B), (D), (F), (H), (J)) for *D. longicolla* isolate DPC-HOH20 ((A) and (B)), *D. caulivora* isolate DPC-HOH2 ((C) and (D)), *D. eres* isolate DPC-HOH7 ((E) and (F)), and *D. novem* isolates DPC-HOH8 ((G) and (H)) and

DPC-HOH11 (I and J).
(TIF)

**S2 Fig. Specificity test for each species-specific TaqMan primer-probe set with *Diaporthe* species.** Specificity test for primer-probe set (A) DPCL, (B) DPCC, (C) DPCE, and (D) DPCN with DNA from *D. longicolla*, *D. caulivora*, *D. eres*, and *D. novem* (from left to right), respectively.
(TIF)

**S3 Fig. Screening soybean seeds via the quadruplex real-time PCR assay.** *D. eres* and *D. novem* were detected in extracted DNA of (A) infected seed coat, (B) infected uncoated seed, and (C) whole infected seed. No amplification was observed for (D) healthy seed coat, (E) healthy uncoated seed, and (F) whole healthy seed.
(TIF)

**S1 Text. Primer BLAST results.**
(DOCX)

**S2 Text. Test of the primer-probe sets in Duplex reactions with both templates.**
(DOCX)

**S1 File. Excel file with multiple datasheets with contents and Cq values from the qPCR experiments reported in this publication.**
(XLSX)

## Acknowledgments

We would like to express our sincere gratitude to Taifun-Tofu GmbH (Freiburg, Germany), to the Landwirtschaftsbetrieb Zschoche (Südliches Anhalt, Germany), and to the Landwirtschaftliches Technologiezentrum (LTZ) Augustenberg (Karlsruhe, Germany) for providing infected and healthy soybean seed lots, and to Daniela Hirschburger for supplying the fungal cultures of *Sclerotinia sclerotiorum*, *Colletotrichum truncatum*, and *Fusarium tricinctum*, and to Kristina Petrović for supplying fungal cultures of *Diaporthe aspalathi*, *Diaporthe foeniculina*, *Diaporthe longicolla*, *Diaporthe eres*, *Fusarium solani*, *Cercospora kikuchii*, and *Alternaria sp.*. We thank Heike Popovitsch for technical assistance.

## Author Contributions

**Conceptualization:** Ralf T. Voegele, Tobias I. Link.

**Funding acquisition:** Behnoush Hosseini, Ralf T. Voegele, Tobias I. Link.

**Investigation:** Behnoush Hosseini, Tobias I. Link.

**Methodology:** Tobias I. Link.

**Project administration:** Tobias I. Link.

**Supervision:** Tobias I. Link.

**Visualization:** Behnoush Hosseini, Tobias I. Link.

**Writing – original draft:** Behnoush Hosseini, Tobias I. Link.

**Writing – review & editing:** Ralf T. Voegele, Tobias I. Link.

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
