## [Decision Letter · Decision Letter 0]

20 Jan 2021

PONE-D-20-38183

Establishment of a quadruplex real-time PCR assay to distinguish the fungal pathogens *Diaporthe longicolla*, *D. caulivora*, *D. eres*, and *D. novem* on soybean

PLOS ONE

Dear Dr. Link,

Thank you for submitting your manuscript to PLOS ONE. After careful consideration, we feel that it has merit but does not fully meet PLOS ONE’s publication criteria as it currently stands. Therefore, we invite you to submit a revised version of the manuscript that addresses the points raised during the review process.

Authors should consider all comments from reviewers who have done a big job to improve this work. There are many comments and they are important, the authors may need to carry out additional experiments in accordance with these recommendations.

We look forward to receiving your revised manuscript.

Kind regards,

Ruslan Kalendar, PhD

Academic Editor

PLOS ONE

Journal Requirements:

Reviewers' comments:

Reviewer's Responses to Questions

**Comments to the Author**

1. Is the manuscript technically sound, and do the data support the conclusions?

Reviewer #1: Partly

Reviewer #2: No

Reviewer #3: Yes

Reviewer #4: Partly

2. Has the statistical analysis been performed appropriately and rigorously? 

Reviewer #1: N/A

Reviewer #2: No

Reviewer #3: Yes

Reviewer #4: Yes

3. Have the authors made all data underlying the findings in their manuscript fully available?

Reviewer #1: No

Reviewer #2: No

Reviewer #3: Yes

Reviewer #4: Yes

4. Is the manuscript presented in an intelligible fashion and written in standard English?

Reviewer #1: Yes

Reviewer #2: Yes

Reviewer #3: Yes

Reviewer #4: Yes

5. Review Comments to the Author

Reviewer #1: 

The manuscript reports the design of primers/probe sets to be used in a multiplex real-time PCR assay to identify four Diaporthe species causing disease on soybean.

First, the authors should state clearly what is the relevance of distinguishing the four different species involved, in terms of disease management. Why do we need to use a costly multiplex real-time PCR assay to help monitoring or preventing the disease? The introduction should explain why the four species should be considered separately and specifically, especially since D. longicolla and D. eres are regarded as the same species according to the current taxonomic databases (see USDA ARS DB). Overall, although I do not question the effort to design specific oligonucleotides, I think that this work lacks a clear and sufficient demonstration of the specificity of the assay. Only a few target species and non-target species have been included in the experiment, and the work does not provide evidence of the application for field samples, and data. Late Ct values observed with non-target species are visible on figure S1, but not discussed at all. A last major issue is the lack of validation of the assay with field samples. Seeds have only been assessed individually, which prevents the use of the test at a practical scale. A protocol to test seed samples (with 400-1000 seeds) should be described. Last, the inhibition and amplifiability of the samples have only been assessed with pure fungal DNA, which has no practical application with field samples. Evaluation of inhibition and amplifiability should be verified with plant DNA, using a reference gene amplification such as COX or 18S rDNA.

I also have other comments that should be addressed to improve the manuscript, listed below. In general, there is too many figures, which should be transferred to supporting material, or simply removed (duplex assay).

I recommend major revisions before acceptance.

L34: remove “anamorph: Phomopsis” since this nomenclature is no longer in force in fungal taxonomy (one fungus, one name)

L71: “providing species specificity”: this part of the sentence does not make sense. Please reword.

L72-74: this sentence referring to reviews is useless here.

L74-L77: why identification of pathogen would provide information on diseases threshold. There is no connection.

Table 1: This is one of the core issue with this study. The shallow number of strains studied and included for each species. Without a significantly higher number of strains, and DNA sequences, there is no possible evaluation of the intra specific polymorphism. Figure 1 shows that such a polymorphism occur for ITS within D. eres. Maybe other intraspecific polymorphism occur for ITS and TEF for other strains of the 4 species, and has been ignored or overlooked. In addition, TEF sequences listed in Table 1 are not available on GenBank. They should be.

L107-112: again, one may regret the very weak number of non-target species included in the specificity assessment (only 3!). Many other species, closely related genetically or also present on Soybean, should be included in the study to support that the multiplex assay does not cross-react with non-target species DNA.

L113: four-month-old

L116: seed lots

L148: how many sequences were used?

L187: “replicates” means how many?

L201: copy numbers are described here, but never used later in the manuscript and in the results section. Why is so?

L246-247: This simply is not true since late Ct values can be observed with a non-target species. Similar late Ct values are observed with individual seed (Fig 10), thus questioning the specificity of the assay or at least interpretation of the results.

L263: Since the goal of the assay is to be used in a multiplex fashion, I do not see the point of assessing the assay in several duplex reactions. Amplifiability and competition should be assessed directly in quadruplex. All the part dealing with duplex should be removed in my opinion.

L275: Competing

L302: Which DNA concentration have been used here?

Discussion: the first part of the discussion L345-383 is just simple repetition of the introduction of or the results and has not added value. The discussion should “discuss” the results, in light of other works, not repeat them.

L395-398: the data showing correlation of isolation and real-time assay are not shown, which is unfortunate. The work should definitely include real-world field samples, not only individual seeds.

L400-L404: please provide information regarding DPC, instead of general seed-borne diseases.

L408-411: this is irrelevant here, it is like presenting the objectives of another work.

L415: the authors should clearly describe why the assay offers a potential to “dramatically” improve lab diagnostic.

Reviewer #2: 

The manuscript, titled “Establishment of a quadruplex real-time PCR assay to distinguish the fungal pathogens Diaporthe longicolla, D. caulivora, D. eres, and D. novem on soybean” developed a singleplex, duplex and quadruplex real-time PCR assay to distinguish four soybean pathogens.

The method could be a detection tool to improve the diagnosis and management of these pathogens. The PCR, especially real-time PCR, could increase specificity and sensitivity compared to more traditional techniques. To achieve this, a reliable method is required. Therefore, a validation study of real-time PCR method should be conducted to confirm by examination and provision of objective evidence that the particular requirements for a specific intended use are fulfilled. The qualitative real-time PCR method must meet acceptance criteria of specificity, sensitivity (limit of detection, LOD), robustness, amplification efficiency and linearity (the latter two optional).

The results of the theoretical specificity test on the BLAST database (Materials and Methods, lines 151-154) are neither shown nor discussed. The experimental sensitivity was tested with target DNA (D. longicolla, caulivora, eres, novem, S1 Fig., Figs 2-9) at unknown concentrations in ng (line 128, “DNA concentrations were determined by measuring the absorption at 260 nm” inconsistent with lines 199-200 “DNA concentrations were determined by using a Qubit 2.0 Fluorimeter” and results are not shown). Determination number for pathogens test is unclear: lines 186-187 “reaction used as standards were run in technical triplicates; reaction used to test for pathogen presence were run in technical replicates”. Non-target DNA tested were S. sclerotiorum DSMZ, S.sclerotiorum IZS, C. truncatum, F. tricinctum, healthy soybean leaf and stem, healthy soybean seed coat, uncoated and whole. Non-target DNA concentrations are unknown and it would be interesting to check specificity with respect to other Diaporthe species such as D. phaseolorum var. sojae, D. phaseolorum var. meridionalis and for the most important related crops.

Lines 102-103 Materials and methods “These Diaporthe isolates were used to test the specificity and sensitivity…”, but the sensitivity was not tested. A test, similar to the asymmetric LOD, was performed for the duplex assays (Table 2), but without first defining a LOD for each target-method and for the proposed quadruplex assay.

Robustness was not performed.

The efficiency results, shown only for single methods (Fig. 2-4, Table 2) and duplex methods (Table 4), are not good for genomic DNA of D. longicolla system (Fig. 3B), D. novem system, isolate HOH11 (Fig. 4D) and duplex PCR, set2 and set 4 (Table 2). Is the efficiency of DPCE, set 1, a transcription error? The amplification efficiency must be between 90 and 110% (-3.6 slope -3.1). The results are partially discussed and justified with the presence of polymerase inhibitors, without verification (241-243, 278, 387-390). Observing the amplification curves of Figs. 5-8 and 10, some systems shown low efficiency, especially D. caulivora.

There are many figures and only one Table 4, reporting Cq, no statistical analysis eg mean, SDr and RSDr have been provided and discussed.

Therefore, the partial specificity and low efficiency for some singleplex and duplex methods are not sufficient to consider the quadruplex method reliable and to support the conclusion.

If not already known and considered useful, I recommend the following document “Guidelines for the single-laboratory validation of qualitative real-time PCR methods-BVL 2016”, but now I do not consider the article to be published.

Reviewer #3: 

The authors created a robust detection system that accurately identified four related species of the genus Diaporthe. By creating a multiplex real-time PCR assay, the authors created a scientifically valuable diagnostic that has the potential to quickly identify four closely related emerging soybean pathogens with lower time and handling costs than current methods. The authors presented clear data that their singleplex assays worked with high levels of PCR efficiency and that the assays did not interact negatively in a multiplex format.

My main concern regards the clarification of certain points of the methodologies used and better explanation of the fungal strains sampled.

Abstract: Please identify how many fungal strains of each species were tested for both the target and non-target fungi

Introduction:

What is the current geographic distribution of these fungi? Are the environmental conditions in central Europe conducive to the spread of the fungus? What are those environmental conditions? Will future predicted climate change patterns make the spread of these fungi more likely?

How does the fungus spread? Is it just through spores being transfered from seed to seed? Is distribution through infected seed lots a current or predicted pathway to spread infection (this particular question may fit better in the conclusions)

Materials and Methods:

Were the stem and leaf samples artificially inoculated with the known strains (that is how I interpreted it but it is unclear)?

Were the seed naturally infected with known or unknown strains? (this I could not determine from the text, but I believe it was known fungal species but unknown strains)?

Were the strains on the naturally infected seeds diagnosed outside of the RT-PCR assay presented? Such as by the culture techniques or classical PCR methods.

What is the known genetic diversity of the tested fungi?

Do the strains tested in vivo represent known diversity?

What is the likelihood that non-target, non-pathogenic fungi could react with the primers and probes? (can be answered better with in silico methods)

The authors used bioinformatic tools to test the primers and probe against the known sequences for the four target fungal species and several non-target species. They need to indicate how many different sequences per species were tested. This will give a better idea of how well the primers and probes work across more strains than were tested in vivo. They should also test, in silico, more non-target fungi that could occur future sampling environments and sister species or sister genera to Diaporthe to test for cross reactions.

Results:

Figures 5 - 7 make better sense as supplemental material

Seed Validation study - Were these done from single seeds or a group of seeds? How are seeds typical screened for the fungus? Did they determine the quantity of fungi present separate from the rt-PCR results? - Can be answered in the methods section

Discussion:

395-396 : The suitability of this assay for detection of Diaporthe species in infected plant material was proven via detection of D. longicolla from stem samples inoculated with this species. - worded too strongly since you only tested one species this way

Reviewer #4: 

Title: Establishment of a quadruplex real-time PCR assay to distinguish the fungal pathogens Diaporthe longicolla, D. caulivora, D. eres, and D. novem on soybean

Manuscript: PONE-D-20-38183

Line 34. “…………..the genus Diaporthe (syn. Phomopsis)…………...”

Line 36. Check the authority name for D. longicolla - Resolving the Diaporthe species occurring on soybean in Croatia (nih.gov)

Line 89. Check authority names for the fungal names.

Lines 113-115. Why were the other species not tested on the stems? Was there seeds inoculated with these four species?

Line 157. How is the PCR product (efficiency in %) > 100 for DPCE/DE? (Table 2).

Line 172. Define higher efficiencies.

Lines 206. The section on results reads more like a discussion. I would suggest reading research papers related to development of qPCR to rewrite this section. For example, there are no results on the primer blast? Or on the cut-off values for the assays.

Lines 237-238. The authors describe that they obtained good efficiencies and lower efficiencies, it would be better to provide what the Ct values look like. There is no information on the cutoff- values of Ct for each of the primer/ probe combination.

Line 281. Table 4. Please list the Cq cut-off values.

Line 330. More information needed on how much Cq value was obtained on the different species on each of the seed samples tested. How did this compare with the traditional isolation method in terms of fungal recovery?

Lines 384-394. I agree with the thoughts, but the main concern is about Diaporthe eres, which is believed to be a complex of species rather than a phylogenetically distinct species. Any thoughts about this? https://core.ac.uk/display/81525550

6. PLOS authors have the option to publish the peer review history of their article (what does this mean?). If published, this will include your full peer review and any attached files.

Reviewer #1: **Yes: **Renaud Ioos

---

## [Author Response · Author response to Decision Letter 0]

18 Aug 2021

Rebuttal Letter

General remarks

We appreciate the effort of both the editor and also the reviewers. With so many reviewers, however, it is not surprising that that some comments were contradictory. Also the overall assessment of our work by the reviewers differed. Nonetheless, we have tried to address all comments below.

When we decided to publish, our plan was to separate primer design and primer testing in singleplex and multiplex reactions from application of the primers to plant and especially field samples, which we wanted to publish separately. Because of the recurrent criticism that our assay is not fully tested yet, we decided to do additional experiments and include additional data concerning primer specificity and quantification.

We acknowledge that our primer-probe-combination for D. eres was not optimal. We have designed a new one. Based on the additional experiments and data considered unnecessary from the first submission we have strongly restructured our paper.

Due to the major revisions needed, additional experiments performed, problems encountered, and other commitments it took us very long to provide this revised manuscript and we want to apologize to all people involved.

Answers to reviewer’s comments

Reviewer#1

The manuscript reports the design of primers/probe sets to be used in a multiplex real-time PCR assay to identify four Diaporthe species causing disease on soybean.

First, the authors should state clearly what is the relevance of distinguishing the four different species involved, in terms of disease management. Why do we need to use a costly multiplex real-time PCR assay to help monitoring or preventing the disease? The introduction should explain why the four species should be considered separately and specifically, especially since D. longicolla and D. eres are regarded as the same species according to the current taxonomic databases (see USDA ARS DB). Overall, although I do not question the effort to design specific oligonucleotides, I think that this work lacks a clear and sufficient demonstration of the specificity of the assay. Only a few target species and non-target species have been included in the experiment, and the work does not provide evidence of the application for field samples, and data. Late Ct values observed with non-target species are visible on figure S1, but not discussed at all. A last major issue is the lack of validation of the assay with field samples. Seeds have only been assessed individually, which prevents the use of the test at a practical scale. A protocol to test seed samples (with 400-1000 seeds) should be described. Last, the inhibition and amplifiability of the samples have only been assessed with pure fungal DNA, which has no practical application with field samples. Evaluation of inhibition and amplifiability should be verified with plant DNA, using a reference gene amplification such as COX or 18S rDNA.

I also have other comments that should be addressed to improve the manuscript, listed below. In general, there is too many figures, which should be transferred to supporting material, or simply removed (duplex assay).

I recommend major revisions before acceptance.

We are planning to study Diaporthe epidemiology in Germany and here we want to distinguish between the species. Beyond our own assessment (Hosseini 2020) no studies determining the prevalent Diaporthe sp. in Germany exist. We now mention this in the introduction. We agree that for standard seed testing a primer probe combination with genus specificity could be more economical.

We were surprised by the reviewer’s statement that D. longicolla and D. eres should be considered as the same species since we recently published on Diaporthe phylogeny ourselves and did not receive any objection against considering these as two different species. We have checked https://nt.ars-grin.gov/fungaldatabases/new_allView.cfm?whichone=Nomenclature&thisName=Diaporthe%20eres&fromallCount=true&organismtype=Fungus and found that the statement of the reviewer is accurate. We, however, disagree with the database entry. Apart from our analysis also the studies of Gomes et al. (2013), Persoonia 31:1-41 and Udayanga et al. (2015), Fungal Biology 119, 383-407, put D. longicolla and D. eres into completely different taxa. There is no indication that these should be considered the same species. Why the database is calling D. eres and D. longicolla the same species we do not understand. The database cites J.M. Santos 2011 for the identity between D. eres and D. longicolla. We cannot follow this argument. We realized that the database is now also citing our own publication. Possibly the database entry should be considered outdated in this case.

We did additional experiments testing our primers against additional non target species. We also did additional experiments regarding quantification. We do not feel that protocols for testing seed samples need to be part of this publication.

We reduced the number of figures.

L34: remove “anamorph: Phomopsis” since this nomenclature is no longer in force in fungal taxonomy (one fungus, one name)

We removed it. The name Phomopsis no longer appears in the manuscript, except in the literature list.

L71: “providing species specificity”: this part of the sentence does not make sense. Please reword.

We rephrased the sentence.

L72-74: this sentence referring to reviews is useless here.

We chose to keep the references illustrating the uses of qPCR but have rephrased the sentence.

L74-L77: why identification of pathogen would provide information on diseases threshold. There is no connection.

We rephrased the sentence.

Table 1: This is one of the core issue with this study. The shallow number of strains studied and included for each species. Without a significantly higher number of strains, and DNA sequences, there is no possible evaluation of the intra specific polymorphism. Figure 1 shows that such a polymorphism occur for ITS within D. eres. Maybe other intraspecific polymorphism occur for ITS and TEF for other strains of the 4 species, and has been ignored or overlooked. In addition, TEF sequences listed in Table 1 are not available on GenBank. They should be.

Based purely on what we described in the manuscript this criticism is justified. However, we have studied polymorphism in Diaporthe in central Europe in our earlier publication [4. Hosseini B, El-Hasan A, Link T, Voegele RT. Analysis of the species spectrum of the Diaporthe/Phomopsis complex in European soybean seeds. Mycological Progress. 2020; 19:455-469. doi: 10.1007/s11557-020-01570-y.]. In this study we did not find any additional intra specific polymorphism that is not covered here. Indeed, the alignment shown in Figure 1 is an excerpt from the alignments used to build the phylogenies shown in [4]. The selected sequences were chosen to represent all polymorphisms. We have added a sentence to the legend of Figure 1 explaining this.

We checked all Accession Numbers for TEF sequences listed in Table 1 using the NCBI search panel and the corresponding sequences were returned.

L107-112: again, one may regret the very weak number of non-target species included in the specificity assessment (only 3!). Many other species, closely related genetically or also present on Soybean, should be included in the study to support that the multiplex assay does not cross-react with non-target species DNA.

To address this issue we tested additional non-target species, including other Diaporthe species (D. aspalathi, D. foeniculina).

L113: four-month-old

Changed.

L116: seed lots

Changed.

L148: how many sequences were used?

Here in Material and Methods we only mention the software and algorithms that were used together with the settings. The concrete data are described in Results. In the corresponding section in the Results we have added a description of which data were used for the alignments.

L187: “replicates” means how many?

“Replicate” was replaced by duplicate.

L201: copy numbers are described here, but never used later in the manuscript and in the results section. Why is so?

These dilution series were performed mainly to determine primer efficiencies. The early idea to also use the standard curves to determine detection limits was discarded as irrelevant for test with real samples. We have now reviewed these data and partially repeated the experiments and do mention an estimated detection limit based on copy numbers.

L246-247: This simply is not true since late Ct values can be observed with a non-target species. Similar late Ct values are observed with individual seed (Fig 10), thus questioning the specificity of the assay or at least interpretation of the results.

We initially thought that it could be ok to ignore these late amplifications. Getting rid of this caused a considerable amount of work. During repeating the experiment, we realized that there were contaminations in some of our materials. After eliminating all contaminations we now have no late amplifications any more. S1 Fig. was redone, it is now S3 Fig. We also replaced parts of Fig 10 (it is now S4 Fig.).

L263: Since the goal of the assay is to be used in a multiplex fashion, I do not see the point of assessing the assay in several duplex reactions. Amplifiability and competition should be assessed directly in quadruplex. All the part dealing with duplex should be removed in my opinion.

To us the duplex experiments seemed important. But we do see the point the referee is making. We moved these results to supplementary information. We found the concentrations in our records. The experiments involving D. eres were redone with the new primer-probe set.

We now included new standard curves designed for quantification of the pathogens for the quadruplex.

L275: Competing

Corrected in Supporting Information.

L302: Which DNA concentration have been used here?

Concentrations of genomic DNA preparations were only determined by spectroscopy and did not seem important for the general message of the experiment. We now mention the range of concentrations in the legend of Fig 2.

Discussion: the first part of the discussion L345-383 is just simple repetition of the introduction of or the results and has not added value. The discussion should “discuss” the results, in light of other works, not repeat them.

This is true. Since our paper deals with the establishment of a method, we found it hard to find issues to discuss. We have now included more text discussing how our assay can help in assessing Diaporthe epidemiology and determine what are the dominant Diaporthe species. We also discuss the difficulties of establishing a multiplex assay.

L395-398: the data showing correlation of isolation and real-time assay are not shown, which is unfortunate. The work should definitely include real-world field samples, not only individual seeds.

We are not quite sure what kind of data should be shown here. A table of the plants that were infected? Pictures of stems or plates with mycelium? We stated our plans for including results with field samples in a later publication at the top of this document.

L400-L404: please provide information regarding DPC, instead of general seed-borne diseases.

L408-411: this is irrelevant here, it is like presenting the objectives of another work.

L415: the authors should clearly describe why the assay offers a potential to “dramatically” improve lab diagnostic.

This part of the discussion was included to explain why this manuscript does not yet include results from field samples.

The word dramatically was removed.

We now also show the results of sampling two seed lots. This is discussed in the light of what additional information our assay provides.

Reviewer #2: 

The manuscript, titled “Establishment of a quadruplex real-time PCR assay to distinguish the fungal pathogens Diaporthe longicolla, D. caulivora, D. eres, and D. novem on soybean” developed a singleplex, duplex and quadruplex real-time PCR assay to distinguish four soybean pathogens.

The method could be a detection tool to improve the diagnosis and management of these pathogens. The PCR, especially real-time PCR, could increase specificity and sensitivity compared to more traditional techniques. To achieve this, a reliable method is required. Therefore, a validation study of real-time PCR method should be conducted to confirm by examination and provision of objective evidence that the particular requirements for a specific intended use are fulfilled. The qualitative real-time PCR method must meet acceptance criteria of specificity, sensitivity (limit of detection, LOD), robustness, amplification efficiency and linearity (the latter two optional).

We thank reviewer #2 for recommending “Guidelines for the single-laboratory validation of qualitative real-time PCR methods-BVL 2016”. Indeed, the document was unknown to us so far. We have read the document and it became clear that reviewer #2 used the criteria formulated there for assessing the manuscript. We do agree that the document is useful and we are aware that there are experiments still to be performed before our assay can be used for certification of soybean seed lots. At the same time, we think that the criteria that were formulated by a committee within the German federal office for consumer protection and food safety (BVL) for assessing methods to detect GM plants before including them in the BVL’s Official Collection of Methods cannot without reservations be applied for publications reporting experimental results. We do want our method to be developed into a standard tool for detection of Diaporthe species in Germany, but this is not our application for registration of this tool yet, rather a report on our current results.

The results of the theoretical specificity test on the BLAST database (Materials and Methods, lines 151-154) are neither shown nor discussed. The experimental sensitivity was tested with target DNA (D. longicolla, caulivora, eres, novem, S1 Fig., Figs 2-9) at unknown concentrations in ng (line 128, “DNA concentrations were determined by measuring the absorption at 260 nm” inconsistent with lines 199-200 “DNA concentrations were determined by using a Qubit 2.0 Fluorimeter” and results are not shown). Determination number for pathogens test is unclear: lines 186-187 “reaction used as standards were run in technical triplicates; reaction used to test for pathogen presence were run in technical replicates”. Non-target DNA tested were S. sclerotiorum DSMZ, S.sclerotiorum IZS, C. truncatum, F. tricinctum, healthy soybean leaf and stem, healthy soybean seed coat, uncoated and whole. Non-target DNA concentrations are unknown and it would be interesting to check specificity with respect to other Diaporthe species such as D. phaseolorum var. sojae, D. phaseolorum var. meridionalis and for the most important related crops.

We repeated the analysis with a wider species range (Diaporthe, Fungi, Phythium, Phytophthora, Glycine). We added a couple of sentences in results to describe that using these settings the primer-probe combinations are perfectly suited for what they are needed for. More details about the primer BLAST results are given in a newly written S1 Text. Some unintended targets might limit the geographic range of our assay or possible detection of two different species has to be accepted. Since the goal of our assay is to distinguish between the four species reported here, there does not seem to be a problem with primer specificity.

DNA concentrations are now reported. “Replicates” was replaced with duplicates or triplicates. Additional species were tested.

Lines 102-103 Materials and methods “These Diaporthe isolates were used to test the specificity and sensitivity…”, but the sensitivity was not tested. A test, similar to the asymmetric LOD, was performed for the duplex assays (Table 2), but without first defining a LOD for each target-method and for the proposed quadruplex assay.

We agree that the missing sensitivity test is a shortcoming. We have performed tests with dilution series of DNA prepared from the isolates with precisely determined concentrations. These can be used for quantifications of the pathogen. From these curves also the sensitivity can be deduced even though we still cannot fully define the LOD.

Robustness was not performed.

The test for robustness should be performed at some instance to register the assay but development of the assay has not yet progressed so far. Robustness should not a criterion for a publication. None of the previous publications on qPCR detection of Diaporthe pathogens features a test for robustness.

The efficiency results, shown only for single methods (Fig. 2-4, Table 2) and duplex methods (Table 4), are not good for genomic DNA of D. longicolla system (Fig. 3B), D. novem system, isolate HOH11 (Fig. 4D) and duplex PCR, set2 and set 4 (Table 2). Is the efficiency of DPCE, set 1, a transcription error? The amplification efficiency must be between 90 and 110% (-3.6 slope -3.1). The results are partially discussed and justified with the presence of polymerase inhibitors, without verification (241-243, 278, 387-390). Observing the amplification curves of Figs. 5-8 and 10, some systems shown low efficiency, especially D. caulivora.

New efficiency tests are provided for part of the primer-probe sets, especially for the new DPCE..

There are many figures and only one Table 4, reporting Cq, no statistical analysis eg mean, SDr and RSDr have been provided and discussed.

Development of our method has not progressed so far yet. We will provide these data in later publications.

Therefore, the partial specificity and low efficiency for some singleplex and duplex methods are not sufficient to consider the quadruplex method reliable and to support the conclusion.

We disagree, see earlier comments. Also, further progress was made in the meantime.

If not already known and considered useful, I recommend the following document “Guidelines for the single-laboratory validation of qualitative real-time PCR methods-BVL 2016”, but now I do not consider the article to be published.

We already commented on this above. We thank reviewer #2 for recommending “Guidelines for the single-laboratory validation of qualitative real-time PCR methods-BVL 2016”. Indeed, the document was unknown to us so far. We do agree that the document is useful but we also think that these criteria should not be applied to a publication. 

Reviewer #3: 

The authors created a robust detection system that accurately identified four related species of the genus Diaporthe. By creating a multiplex real-time PCR assay, the authors created a scientifically valuable diagnostic that has the potential to quickly identify four closely related emerging soybean pathogens with lower time and handling costs than current methods. The authors presented clear data that their singleplex assays worked with high levels of PCR efficiency and that the assays did not interact negatively in a multiplex format.

We thank reviewer #3 for this overall very positive assessment of our study.

My main concern regards the clarification of certain points of the methodologies used and better explanation of the fungal strains sampled.

Abstract: Please identify how many fungal strains of each species were tested for both the target and non-target fungi

We considered adding this information to the abstract but found that it would add considerably to the length of the abstract and decided that this information is not so highly relevant that it should be represented in the abstract.

Introduction:

What is the current geographic distribution of these fungi? Are the environmental conditions in central Europe conducive to the spread of the fungus? What are those environmental conditions? Will future predicted climate change patterns make the spread of these fungi more likely?

All four species have previously been identified in southern and south eastern Europe. We now refer to this in the discussion. Mostly we rely on our own identification of the species in central Europe. Mostly so far the species are limited in their spread throughout central Europe because there is little soybean so far. Spread of the fungi in central Europe will, therefore, be accelerated mainly by the increase in soybean production.

How does the fungus spread? Is it just through spores being transfered from seed to seed? Is distribution through infected seed lots a current or predicted pathway to spread infection (this particular question may fit better in the conclusions)

Indeed, the fungi grow into the seeds and infected seeds lead to infected new plants. Apart from that there is sporulation on the stems and infected plant residues also are a major source of inoculum. It is now mentioned in the introduction that research is necessary regarding these questions and that our assay will help in answering them.

Materials and Methods:

Were the stem and leaf samples artificially inoculated with the known strains (that is how I interpreted it but it is unclear)?

It is somewhat unfortunate that the reviewer did not provide line numbers. It is not fully clear what the questions refer to. Not sure if our answers meet with the questions. – Yes, the samples were inoculated with the known strains mentioned in Table 1. We changed a few words to make this more clear.

Were the seed naturally infected with known or unknown strains? (this I could not determine from the text, but I believe it was known fungal species but unknown strains)?

Were the strains on the naturally infected seeds diagnosed outside of the RT-PCR assay presented? Such as by the culture techniques or classical PCR methods.

This probably refers to l116ff. We think in this case that a broader explanation than what is provided is not necessary. Since our method of preparing DNA from seeds is destructive, diagnosis with culture techniques is not possible. Classical PCR would not be a true corroboration since it depends on the same principle. So no, no diagnosis outside qPCR. However, we used the same samples as used in our earlier study identifying the strains, so these results should serve as an independent diagnosis.

What is the known genetic diversity of the tested fungi?

Do the strains tested in vivo represent known diversity?

We assumed that our isolates from [4] represent the genetic diversity of the tested fungi in central Europe. This diversity was represented by the strains tested.

What is the likelihood that non-target, non-pathogenic fungi could react with the primers and probes? (can be answered better with in silico methods)

This likelihood is very small. We did a new primer BLAST. This is now described in S1 text. Our claim that our assay distinguishes between D. caulivora, D. eres, D. longicolla, and D. novem was fully corroborated.

The authors used bioinformatic tools to test the primers and probe against the known sequences for the four target fungal species and several non-target species. They need to indicate how many different sequences per species were tested. This will give a better idea of how well the primers and probes work across more strains than were tested in vivo. They should also test, in silico, more non-target fungi that could occur future sampling environments and sister species or sister genera to Diaporthe to test for cross reactions.

We changed the wording in results (l220ff) explaining this. We initially included all sequences from our earlier phylogenetic study but then removed all identical sequences. We hope that our new wording explains this properly.

Results:

Figures 5 - 7 make better sense as supplemental material

We changed several figures. We reduced the number of figures and put some into the supplementary material. Figures 5 -7 were made into one. We hope that this meets with this comment.

Seed Validation study - Were these done from single seeds or a group of seeds? How are seeds typical screened for the fungus? Did they determine the quantity of fungi present separate from the rt-PCR results? - Can be answered in the methods section

This was done with single seeds. The full seed screening methodology still needs to be established. We have now added extra results showing testing of two seed samples and also added the calculated quantification.

Discussion:

395-396 : The suitability of this assay for detection of Diaporthe species in infected plant material was proven via detection of D. longicolla from stem samples inoculated with this species. - worded too strongly since you only tested one species this way

Changed the wording from proven to supported.

Reviewer #4: 

Title: Establishment of a quadruplex real-time PCR assay to distinguish the fungal pathogens Diaporthe longicolla, D. caulivora, D. eres, and D. novem on soybean

Manuscript: PONE-D-20-38183

Line 34. “…………..the genus Diaporthe (syn. Phomopsis)…………...”

According to the recommendation of reviewer #1 we removed “anamorph Phomopsis”.

Line 36. Check the authority name for D. longicolla - Resolving the Diaporthe species occurring on soybean in Croatia (nih.gov)

Line 89. Check authority names for the fungal names.

Did so. As to our current knowledge, these are all correct. 

Lines 113-115. Why were the other species not tested on the stems? Was there seeds inoculated with these four species?

Only stems inoculated with D. longicolla were available. Since seeds were available with natural infections for all four species no artificial inoculations were performed.

Line 157. How is the PCR product (efficiency in %) > 100 for DPCE/DE? (Table 2).

We were not sure either. Question is no longer relevant since the DPCE primer probe set was replaced.

Line 172. Define higher efficiencies.

101.7 % or 98.3 % respectively as shown in Table 2, now also irrelevant because the DPCE primer probe set was replaced.

Lines 206. The section on results reads more like a discussion. I would suggest reading research papers related to development of qPCR to rewrite this section. For example, there are no results on the primer blast? Or on the cut-off values for the assays.

Part of this is very general criticism. We hope that with all the changes made our results section reads better now. We have added a description of the primer BLAST results. We did additional experiments and have added standard curves for the quadruplex reaction that allow for quantification of the Diaporthe species in a given sample. From these standard curves also detection limits and cut-off values can be deduced, even though the exact LOD still needs to be determined.

Lines 237-238. The authors describe that they obtained good efficiencies and lower efficiencies, it would be better to provide what the Ct values look like. There is no information on the cutoff- values of Ct for each of the primer/ probe combination.

See comments above. Describing our experiments we did find it suitable to mention the results of our primer efficiency tests.

Line 281. Table 4. Please list the Cq cut-off values.

We have done an estimate for Cq cut-off values. 

Line 330. More information needed on how much Cq value was obtained on the different species on each of the seed samples tested. How did this compare with the traditional isolation method in terms of fungal recovery?

We now show an example for testing a full seed sample. A comparison for recovery from single seeds cannot be done, since the seed is destroyed by DNA-preparation and cannot be used for the traditional method. Using the traditional method on a full seed sample was not yet performed.

Lines 384-394. I agree with the thoughts, but the main concern is about Diaporthe eres, which is believed to be a complex of species rather than a phylogenetically distinct species. Any thoughts about this? https://core.ac.uk/display/81525550

It cannot be denied that the taxonomic status of D. eres is ambiguous. However, our own results (Hosseini 2020) indicate that in Central Europe (Germany) we are dealing with a single species. In Hosseini 2020 we also discuss the taxonomic issues of all four species tested here. We feel that a discussion on taxonomic issues is not called for in this paper dealing with diagnostics.

---

## [Decision Letter · Decision Letter 1]

27 Aug 2021

Establishment of a quadruplex real-time PCR assay to distinguish the fungal pathogens *Diaporthe longicolla*, *D. caulivora*, *D. eres*, and *D. novem* on soybean

PONE-D-20-38183R1

Dear Dr. Link,

We’re pleased to inform you that your manuscript has been judged scientifically suitable for publication and will be formally accepted for publication once it meets all outstanding technical requirements.

Kind regards,

Ruslan Kalendar

Academic Editor

PLOS ONE

Reviewers' comments:

Reviewer #1:

Just a few minor editorial comments left (based on track change version numbering):

L104: affiliation of Mrs Pertrovic should be added

L224: I suggest to remove this sentence, which is useless

Table 2: the sequence of the reverse primers is usually written in the 5' - 3' direction.

L427: should read SYBR

---

## [Editor Report · Acceptance letter]

2 Sep 2021

PONE-D-20-38183R1 

Establishment of a quadruplex real-time PCR assay to distinguish the fungal pathogens *Diaporthe longicolla, D. caulivora, D. eres*, and *D. novem* on soybean 

Dear Dr. Link:

I'm pleased to inform you that your manuscript has been deemed suitable for publication in PLOS ONE. Congratulations! Your manuscript is now with our production department. 

Kind regards, 

on behalf of

Professor Ruslan Kalendar 

Academic Editor

PLOS ONE